# MET: Masked Encoding for Tabular Data

## Abstract

We propose *Masked Encoding for Tabular Data (MET)* for learning self-supervised representations from *tabular data*. Tabular self-supervised learning (tabular-SSL) – unlike structured domains like images, audio, text – is more challenging since each tabular dataset can have a completely different structure among its features (or coordinates), which is hard to identify a priori. MET attempts to circumvent this problem by assuming the following hypothesis: the observed tabular data features come from a latent graphical model and the downstream tasks are significantly easier to solve in the latent space. Based on this hypothesis, MET uses random masking based encoders to learn a positional embedding for each coordinate, which would in turn capture the latent structure between coordinates. Through experiments on a toy dataset from a linear graphical model, we show that MET is indeed able to capture the latent graphical model. Practically, through extensive experiments on multiple benchmarks for tabular data, we demonstrate that MET significantly outperforms all the baselines. For example, on Criteo – a large-scale click prediction dataset – MET achieves as much as $5\%$ improvement over the current state-of-the-art (SOTA) while purely supervised learning based approaches have been able to advance SOTA by at most $2\%$ in the last six years. Furthermore, averaged over *nine* datasets, MET is around $3.9\%$ more accurate than the next best method of Gradient-boosted decision trees – considered as SOTA for the tabular setting.

## 1 Introduction

Self-supervised pre-training (SSL) followed by supervised fine-tuning has emerged as the state of the art approach for multiple domains such as natural language processing (NLP) (Devlin et al., 2019), computer vision (Chen et al., 2020b) and speech/audio processing (Baevski et al., 2020). However, despite presence of an extensive amount of raw and unlabeled data in a variety of critical tabular-heavy domains like finance, marketing, etc., it has been challenging to extend SSL based pre-training approaches to tabular data.

Broadly speaking, there are two dominant approaches to SSL: (i) reconstruction of masked inputs, and (ii) invariance to certain augmentations/transformations, also known as *contrastive learning*. Most of the existing tabular-SSL methods (Verma et al., 2020; Ucar et al., 2021) have adopted the second approach of contrastive learning. The underlying structure and semantics of specific domains such as images remain somewhat static, irrespective of the dataset. So, one can design generalizable domain specific augmentations like cropping, rotating, resizing etc. However, tabular data does not have such fixed input vocabulary space (such as pixels in images) and semantic structure, and thus lacks generalizable augmentations across different datasets. Consequently, there are only a limited number of augmentations that have been proposed for the tabular setting such as mix-up, adding random (gaussian) noise and selecting subsets of features (Verma et al., 2020; Ucar et al., 2021).

In this paper, we hypothesize the following: for any tabular dataset, (i) there is a latent (i.e., unknown/unobserved) graphical model that captures the relations between different coordinates/features, and (ii) classification is easier in the latent space. For example, in the CovType dataset – where the task is to predict the type of forest (e.g., deciduous, alpine etc.) given features such as elevation, soil type – extensive research in mountain and forest science has established that there are very specific relations among different features (Catry et al., 2009; Badía et al., 2016), and leveraging and learning these relations could yield significant improvements in classification accuracy of machine learning models.

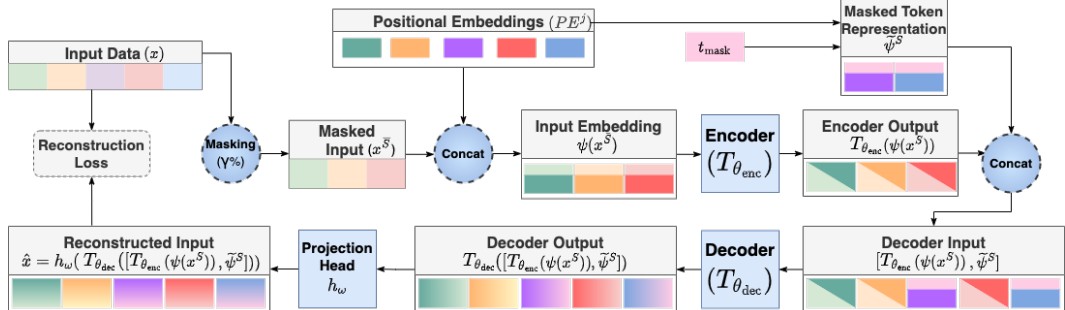

**Figure 1:** MET Framework for tabular-SSL. Given an input $x$, we mask $\gamma$ fraction of it's coordinates to get the masked input $x^{\bar{S}}$. We then concatenate the coordinate value $x^j$ with a learnable positional encoding $PE^j$ to obtain its input embedding $\psi(x^{\bar{S}})$. The obtained input embedding is then passed through the encoder $T_{\theta_{enc}}$. The encoder output $T_{\theta_{enc}}(\psi(x^{\bar{S}}))$ is then concatenated with the masked token representations $\widetilde{\psi}^S$, and given as the input to the decoder $T_{\theta_{dec}}$. Finally, the decoder output is projected back to the input space using a projection head $h_\omega$, to obtain the reconstructed input $\hat{x}$. Reconstruction loss is then optimized end-to-end.

Based on this hypothesis, we propose a masking-based reconstruction approach for self-supervised learning for tabular datasets. More concretely, for every unlabeled data point, we randomly choose a fraction of the coordinates, mask their values, and then train a model to predict the values of these masked coordinates using the remaining unmasked coordinates. We use a transformer architecture with learnable (positional) embeddings for each coordinate, which capture the relations between different coordinates. While masked reconstruction task with a transformer architecture has been successfully used for SSL in computer vision (He et al., 2021) and NLP (Devlin et al., 2019), to the best of our knowledge, this is the first work to successfully apply this paradigm to tabular datasets.

In particular, we demonstrate through experiments on a simple toy tabular setting, how the position embeddings in a transformer, learned with masked reconstruction task, can capture the dependency structure across features. Further, on a real-world dataset of forest cover type classification, we show that features with the most correlated positional embeddings indeed have meaningful relations between them, as corroborated by extensive works in the forest science literature.

We evaluate the performance of MET through extensive experiments on *nine* datasets spanning a wide range in number of examples, classes and difficulty. Our experiments show that MET outperforms current SOTA tabular-SSL methods like DACL (Verma et al., 2020), SubTab (Ucar et al., 2021), VIME (Yoon et al., 2020), as well as SOTA tabular supervised algorithms such as gradient boosted decision trees (GBDT) (Mason et al., 1999) on *all* of these datasets. MET gives an average accuracy improvement of 3.9% over the second best algorithm, which is GBDT. For example, on Criteo – a popular large scale dataset for click through rate prediction with 45 million examples – MET achieves 5% improvement in AUROC over the current SOTA (Wang et al., 2021). To put this in perspective, the SOTA on Criteo has improved by less than 2% over the last six years Leaderboard (2022). Furthermore, on some datasets, MET trained with about 20% of the labeled train-set is as effective as standard supervised learning methods trained with all the labeled points in the train-set.

To summarize, in this paper, we propose MET, which is a masking based reconstruction approach with a transformer architecture, and demonstrate its effectiveness for tabular-SSL. Conceptually, through experiments on a toy setting, we show that this approach can learn the relations between different coordinates in the dataset, which helps in downstream classification. Practically, we show through extensive experiments on several popular tabular datasets that MET significantly outperforms all the current SOTA tabular-SSL baselines as well as SOTA supervised approaches.

## 2 RELATED WORK

**Self-Supervised Learning (SSL) :** SSL has shown promising results not only in the regimes where the labelled training data is scarce but has also shown great empirical success in training large-scale models across various domains like Natural Language Processing and Computer Vision. SSL can be broadly classified into two categories: Pretext task based approaches and contrastive learning based approaches. Pretext based SSL approaches solve a "pretext" task like reconstruction from a masked or a noisy input, to learn the underlying distribution of the unlabeled data.

Wav2Vec(Baevski et al., 2020) efficiently trains a large scale speech-to-text model using masked reconstruction. Similarly, Jiang et al. (2019); Hsu et al. (2021) have used masking for learning speech representations. Masked language modelling has been extensively studied in literature (Devlin et al., 2019; Radford & Narasimhan, 2018), and has shown quite promising results. Motivated by the success of masking based approaches, a recent paper He et al. (2021) proposed a similar approach for visual representation learning. Prior to this, Li et al. (2021) also proposed masking, although in feature space. In this paper, motivated by the constraints in tabular setting along with the success of masking based approaches, we build a purely reconstruction based approach for tabular-SSL.

A concurrent line of work in SSL learns representations using instance level separation tasks as discussed in Chen et al. (2020b); Chen & He (2021); Zbontar et al. (2021); He et al. (2020). However, such approaches require domain knowledge to create positive-negative sample pairs. Recently, domain agnostic augmentations for creating separation tasks have been proposed (Verma et al., 2020). We compare MET against such approaches in our empirical study (Section 5).

**SSL for Tabular Data**   Reconstruction based SSL has been previously explored in SubTab Ucar et al. (2021), which treats it as a multi-view representation learning problem. They try to learn representations for multiple croppings of the input data and at inference time aggregate the representations of the croppings (multiple-views). Note that MET performs random masking over the input space only during training, at inference the representation is given by passing all the coordinates through the encoder and hence does not consider the problem as a multi-view representation learning. Yoon et al. (2020) uses a combination of predicting the masked tokens and reconstruction. Both Yoon et al. (2020); Ucar et al. (2021) add gaussian noise to prevent indentity map learning by the auto-encoder. MET efficiently searches for noise using adversarial search to learn better representations. Another work Huang et al. (2020) uses transformers to solve a pretraining task of predicting the class of missing categorical features using cross-entropy. However, MET uses masked reconstruction and doesn't make any assumption on the type of features i.e. categorical or continuous.

**Adversarial Self Supervised Learning**   While adversarial SSL has been explored in the context of contrastive learning (Kim et al., 2020), it seems to be less explored for reconstruction based SSL. Our method MET proposes a novel framework where we try to find adversarial points in the input manifold which have a high reconstruction loss. Chen et al. (2020a) have also proposed adversarial learning, although to learn robust pre-trained models. MET instead explores the use of adversarial search over input manifold to learn better separable representations for higher accuracy on downstream classification. Shi et al. (2022) proposes to find an adversarial mask which maximizes the distance between the representations of input and its masked (adversarial) counterpart.

**Learning Graphical Models**   : There is a large body of work on unsupervised learning of probabilistic relations between different coordinates/features, as expressed by a graphical model (Chow & Liu, 1968; Bach & Jordan, 2002; Bresler et al., 2008; Netrapalli et al., 2010; Klivans & Meka, 2017). A popular approach in these works is to predict the value of any single coordinate using all the remaining coordinates. This can however be computationally challenging since this procedure needs to be repeated for every coordinate. The masking based reconstruction task with a random subset of coordinates masked in every step can be seen as a computationally efficient way of doing the same thing. To the best of our knowledge, this is the first work to connect masking based reconstruction approaches for SSL with the work on learning graphical models.

## 3  PRELIMINARIES

**Notation:** We use $x \in \mathbb{R}^d$ to denote an input datapoint and $x^j$ to denote the $j^{\text{th}}$ coordinate of $x$. Every coordinate in $x$ i.e. $x^j \in \mathbb{R}$ can be either a categorical or a non-categorical value, without being explicitly specified. Let $S$ denote the set of masked co-ordinates for $x$ and $\bar{S} = [d] \setminus S$ be the set of non-masked coordinates. $S$ is constructed by randomly sampling $\lfloor \gamma d \rfloor$ coordinates from the set $[d]$, where $\gamma$ is the masking ratio hyper-parameter. Note that we sample a different $S$ in every training iteration. Let $x^{\bar{S}}$ denote the masked input, i.e. $x^{\bar{S}}$ consists of only the non-masked coordinates/features $x^j, \forall j \in \bar{S}$.

**Self-Supervised Representation Learning:** Consider access to a corpus of unlabelled data given by $\mathcal{D}_u = \{x_i\}_{i=1}^{N_u}$ where each input $x_i \in \mathbb{R}^d$. The general goal of SSL is to learn a parameterized

mapping $T_{\theta_{\text{enc}}} : \mathbb{R}^d \to \mathbb{R}^m$ between the input $x_i$ and its representation $T_{\theta_{\text{enc}}}(x_i) \in \mathbb{R}^m$, such that the representations are well suited for a downstream task as described next.

**Evaluation of Learned Representations:** In this paper, we evaluate the quality of learned representations through accuracy on a downstream classification task. More concretely, we have access to a labelled training dataset $\mathcal{D}_{\text{train}} = \{(x_i, y_i)\}_{i=1}^{N_{\text{train}}}$ where $y_i \in \mathbb{R}^k$ and each $(x_i, y_i)$ is drawn independently and identically (i.i.d.) from some underlying distribution $\mathcal{D}$ on $\mathbb{R}^d \times \mathbb{R}^k$. The task is to learn a classifier $c_\phi : \mathbb{R}^d \to \mathbb{R}^k$ which minimizes $\mathbb{E}_{(x,y) \sim \mathcal{D}} [\ell(c_\phi(x), y)]$, where $\ell$ is a loss function such as $0 - 1$ loss or cross entropy loss etc. Given the learned encoder network $T_{\theta_{\text{enc}}} : \mathbb{R}^d \to \mathbb{R}^m$, we train a shallow classifier $g_\mu : \mathbb{R}^m \to \mathbb{R}^k$ (we use a 2-hidden layer MLP in our default setting) and use the resulting accuracy to evaluate the quality of learned representations from $T_{\theta_{\text{enc}}}$ i.e., we set $c_\phi(x) = g_\mu(T_{\theta_{\text{enc}}}(x))$ and train only $g_\mu$.

## 4 METHOD

As motivated in Section 1, our proposed approach MET is based on the hypothesis that there is an underlying graphical model which captures the relation between the various coordinates (features) of the tabular data. MET tries to learn this graphical model, using a masked reconstruction based technique. More concretely, this generally involves having an encoder $T_{\theta_{\text{enc}}} : \mathbb{R}^d \to \mathbb{R}^m$ and a decoder $T_{\theta_{\text{dec}}} : \mathbb{R}^m \to \mathbb{R}^d$. Encoder's task is to take a noisy version of the input $x$ (e.g., where some coordinates of $x$ are masked) and map it to some latent representations $\Phi(x)$. The decoder then maps the representations back to the input space to get a reconstructed input $\hat{x}$. The reconstruction error $||x - \hat{x}||_2^2$ is then generally used as a learning objective (although MET also involves an adversarial objective as explained later). Note that this masked reconstruction task, interpreted through the lens of learning graphical models, can learn the underlying latent structure among different coordinates (Klivans & Meka, 2017), thereby making downstream classification tasks easier.

Our proposed model to get the reconstructed input $\hat{x}$ is given by :

$$\hat{x} := h_\omega(T_{\theta_{\text{dec}}}([T_{\theta_{\text{enc}}}(\psi(x^{\bar{S}})), \widetilde{\psi}^S])),$$
$$s.t., \quad [\psi(x^{\bar{S}})]^j = \text{Concat}(x^j, PE^j), \quad \forall j \in \bar{S}, \quad [\widetilde{\psi}^S]^j = \text{Concat}(t_{\text{mask}}, PE^j), \quad \forall j \in S. \quad (1)$$

Parsing the above Equation 1, given an input $x$ and its set of masked coordinates $S$, we obtain its input embedding $\psi(x^{\bar{S}})$ by concatenating the coordinate value $x^j$ with a learnable positional encoding $PE^j$ for every *unmasked* coordinate $j \in \bar{S}$. The obtained input embedding is then passed through the encoder $T_{\theta_{\text{enc}}}$. The encoder output $T_{\theta_{\text{enc}}}(\psi(x^{\bar{S}}))$ is then concatenated with the masked token representations $\widetilde{\psi}^S$, and given as input to the decoder $T_{\theta_{\text{dec}}}$. Finally, the decoder output is projected back to the input space using a projection head $h_\omega$, to obtain the reconstructed input $\hat{x}$.
**We explain each of the components in detail below**:

1. **Input Embedding** : Given an input $x$ and it's set of masked coordinates $S$, recall that $x^{\bar{S}}$ denotes the masked input. Then $\psi(x^{\bar{S}}) \in \mathbb{R}^{|\bar{S}| \times d_{\text{embed}}}$ denotes the input embedding corresponding to the masked input. It is constructed in a per-coordinate fashion by concatenating the coordinate value $x^j \in \mathbb{R}$ with a learnable positional encoding $PE^j \in \mathbb{R}^{d_{\text{embed}}-1}, \forall j \in \bar{S}$.

$$[\psi(x^{\bar{S}})]^j = \text{Concat}(x^j, PE^j) \in \mathbb{R}^{d_{\text{embed}}}, \forall j \in \bar{S}. \quad (2)$$

2. **Encoder** : Given that transformers (Vaswani et al., 2017b) explicitly capture the relation between different coordinates through positional embeddings and attention mechanism, they are a natural choice for the encoder $T_{\theta_{\text{enc}}}$ and the decoder $T_{\theta_{\text{dec}}}$ architecture. Later in Section 4.3, we show that the learnt positional embeddings $PE$ in MET, indeed capture the underlying graphical model, indicating why MET is so successful on various downstream tasks.

   The representations for the masked input i.e. the encoder output is now given by:

$$T_{\theta_{\text{enc}}}(\psi(x^{\bar{S}})) \in \mathbb{R}^{|\bar{S}| \times d_{\text{embed}}}. \quad (3)$$

3. **Masked Token Representations** : Before we pass the encoder output to the decoder, we concatenate it with the masked token representations i.e. the representations for the coordinates which

were masked, given by $\widetilde{\psi}^S \in \mathbb{R}^{|S| \times d_{\mathrm{embed}}}$. Again, $\widetilde{\psi}^S$ is constructed in a per-coordinate fashion, by concatenating a fixed learnable "mask token" $t_{\mathrm{mask}} \in \mathbb{R}$ with $PE^j$.

$$[\widetilde{\psi}^S]^j = \mathrm{Concat}(t_{\mathrm{mask}}, PE^j) \in \mathbb{R}^{d_{\mathrm{embed}}}, \forall j \in S. \tag{4}$$

Note that we do not pass the masked token representations $\widetilde{\psi}^S$ through the encoder.

4. **Decoder And Projection Head** : The concatenated encoder output and masked token representation i.e. $[T_{\theta_{\mathrm{enc}}}(\psi(x^{\bar{S}})), \widetilde{\psi}^S]$ is passed as input to the decoder $T_{\theta_{\mathrm{dec}}}$. Finally, the decoder output is mapped back to the input space using a linear projection head $h_\omega : \mathbb{R}^{d \times d_{\mathrm{embed}}} \to \mathbb{R}^d$, to obtain the reconstructed input $\hat{x}$ as given by Equation 1.

### 4.1 MET-S : Only reconstruction based objective

First, a simplified version of our proposed approach, which we call MET-S, uses just the reconstruction error as the learning objective. Given a corpus of unlabeled dataset $\mathcal{D}_u = \{x_i\}_{i=1}^{N_u}$, the learning objective for MET-S is given by:

$$\mathcal{L}_{\mathrm{MET\text{-}S}}(\theta_{\mathrm{enc}}, \theta_{\mathrm{dec}}, \omega, PE, t_{\mathrm{mask}}) = \sum_{i=1}^{N_u} \|x_i - \hat{x}_i\|_2^2., \tag{5}$$

where $\hat{x}_i$ denotes the reconstructed input for $x_i$, as given by Equation 1.

At inference time (downstream classification), we don't mask the input i.e. $S = \{\}$ and $\bar{S} = [d]$. Also, we drop the decoder $T_{\theta_{\mathrm{dec}}}$ and simply use the "flattened" encoder output as learnt representations. More concretely, given $x \in \mathbb{R}^d$, we pass the whole input through the encoder and "flatten" the encoder output to get the learnt representations $\Phi(x) = T_{\theta_{\mathrm{enc}}}(\psi(x^{[d]})) \in \mathbb{R}^m$ where $m = d \cdot d_{\mathrm{embed}}$.

### 4.2 MET: Adversarial reconstruction

In the context of supervised learning, several papers have demonstrated that adversarial training can yield more robust features (Tsipras et al., 2018) that are better for transfer learning (Salman et al., 2020). While an adversarial loss function has been observed to encourage learning of robust features in contrastive SSL (Chen et al., 2020a; Kim et al., 2020), to the best of our knowledge, it does not seem to have been explored in the context of masked reconstruction. In this work, we demonstrate that an adversarial version of the reconstruction loss works better than standard reconstruction loss. More concretely, the adversarial reconstruction loss is given by:

$$\mathcal{L}_{\mathrm{rec}}^{\mathrm{adv}}(\theta_{\mathrm{enc}}, \theta_{\mathrm{dec}}, \omega, PE, t_{\mathrm{mask}}) = \sum_{i=1}^{N_u} \max_{\delta : \|\delta\| \leq \epsilon} \|x_i - \hat{x}_i^\delta\|_2^2, \tag{6}$$

where $\hat{x}_i^\delta$ denotes the reconstruction corresponding to a perturbed input $x_i + \delta$ in Equation 1 (replacing $x$ with $x + \delta$ in Equation 1). In this paper, we constrain the adversarial noise $\delta$ in an $\epsilon$ radius $L2$ norm ball around the input data point $x_i$, where $\epsilon$ is chosen from a grid-search in $\{2, 4, 6, 10, 12, 14\}$.

Finally, MET minimizes the sum of loss functions given in Eqn. (5) and Eqn. (6) :

$$\mathcal{L}_{\mathrm{rec}}^{\mathrm{MET}} = \mathcal{L}_{\mathrm{MET\text{-}S}} + \lambda \mathcal{L}_{\mathrm{rec}}^{\mathrm{adv}}. \tag{7}$$

We fix $\lambda = 1$ in all our experiments. The overall algorithm for MET, which minimizes Eqn. (7) is given in Algorithm 1. For consistency of notation, we present a non-batch version of the algorithm.

### 4.3 Analysis on Toy Dataset

As explained previously in Section 1, our proposed approach MET relies on learning the graphical model that defines the relation between various features (coordinates) of the tabular dataset. We first show that MET is indeed able to learn this latent graphical model by presenting some interesting results on a 10-dimensional toy dataset.

Let each datapoint $x \in \mathbb{R}^{10}$ be sampled from a linear graphical model as follows (recall that $x^j \in \mathbb{R}$ denotes the $j^{th}$ coordinate of $x$) :

$$x^0 \sim \mathcal{N}(0, 1); \ x^j = (x^{j-1} + n_j)/\sqrt{2}, \forall j \in [1, 9]; \ \text{where } n_j \sim \mathcal{N}(0, 1) \tag{8}$$

---

**Algorithm 1:** MET : Masked Encoding Tabular data

---

**Input** : Unlabelled data $\mathcal{D}_u = \{x_i\}_{i=1}^{N_u}$, masking ratio $\gamma \in [0, 1]$, $N$ iterations, Encoder $T_{\theta_{\text{enc}}}$, Decoder $T_{\theta_{\text{dec}}}$, mask token $t_{\text{mask}}$, projection head $h_\omega$, positional encodings PE, data embedding $\psi(x^{\bar{S}})$, mask token embedding $\widetilde{\psi}^S$, projection radius $\epsilon$, weight of adversarial loss $\lambda$

**for** $iteration = 0, 1, \ldots N-1$ **do**

    **for** $x \in \mathcal{D}_u$ **do**

        $S \subset [d]$ s.t. $|S| = \lfloor \gamma d \rfloor$   `/* Random subset of coordinates to mask */`

        $\hat{x} = h_\omega(T_{\theta_{\text{dec}}}[T_{\theta_{\text{enc}}}(\psi(x^{\bar{S}})), \widetilde{\psi}^S])$   `/* Reconstruct from masked input */`

        $\mathcal{L}_{\text{rec}}^{\text{std}} = \|x - \hat{x}\|_2^2$            `/* Standard reconstruction loss. */`

        $\delta \sim \mathcal{N}(0, \mathcal{I}_d)/\sqrt{d}$     `/* Initialize adversarial perturbation. */`

        **for** $steps$ $in$ $1, 2, \ldots adv\_steps$ **do**

            `/* Find adversarial perturbation` $\delta$ `to maximize`
                `reconstruction loss using gradient ascent. */`

            $\hat{x}^\delta = h_\omega(T_{\theta_{\text{dec}}}[T_{\theta_{\text{enc}}}(\psi((x+\delta)^{\bar{S}})), \widetilde{\psi}^S])$

            $\mathcal{L}_{\text{rec}}(\delta) = \|x - \hat{x}^\delta\|_2^2$

            $\delta = \delta + \eta \frac{\nabla_\delta \mathcal{L}_{\text{rec}}}{\|\nabla_\delta \mathcal{L}_{\text{rec}}\|}$

            $\delta = \frac{\delta}{\|\delta\|} \alpha$ where $\alpha = \|\delta\| \cdot \mathbb{1}[\|\delta\| < \epsilon] + \epsilon \cdot \mathbb{1}[\|\delta\| \geq \epsilon]$

        $\hat{x}^\delta = h_\omega(T_{\theta_{\text{dec}}}[T_{\theta_{\text{enc}}}(\psi((x+\delta)^{\bar{S}})), \widetilde{\psi}^S])$

        $\mathcal{L}_{\text{rec}}^{\text{adv}} = \|x - \hat{x}^\delta\|_2^2$        `/* Adversarial reconstruction loss. */`

        $\mathcal{L}_{\text{total}} = \mathcal{L}_{\text{rec}}^{\text{std}} + \lambda \cdot \mathcal{L}_{\text{rec}}^{\text{adv}}$      `/* Final loss is a sum of standard and adversarial reconstruction losses. */`

        $\Omega = \Omega - \eta \nabla_\Omega \mathcal{L}_{\text{total}}$, where $\Omega = \{\theta_{\text{enc}}, \theta_{\text{dec}}, \omega, t_{\text{mask}}, PE\}$ `/* Gradient descent on the set of all learnable parameters` $\Omega$. `*/`

---

Note that each coordinate $x^j$ is distributed according to $\mathcal{N}(0, 1)$. For downstream classification, the label is given by $y = \mathbb{1}[\sum_{i=1}^9 n_i^2 \geq \Phi^{-1}(0.5)]$ where $\Phi^{-1}$ denotes the inverse CDF for a chi-square distribution with 9 degrees of freedom. This choice is so that we have a balanced set of positive and negative examples.

First, on the downstream task, our proposed approach MET (96.7 accuracy) outperforms all the baselines, most competitive being Gradient Boosted Decision Trees (88.4 accuracy); see Appendix A.4. Furthermore, MET is able to identify the relation between consecutive coordinates and hence the entire underlying graphical model, as demonstrated by the high cosine similarity scores between learned positional embeddings of consecutive coordinates in Figure 2. As the downstream task is fairly simple once the graphical model is learned (i.e. in $n_i$ space), MET is able to provide a significantly more accurate solution. In Section 5.4, we make a similar observation on real world tabular datasets, where MET is able to identify the meaningful relations among all features.

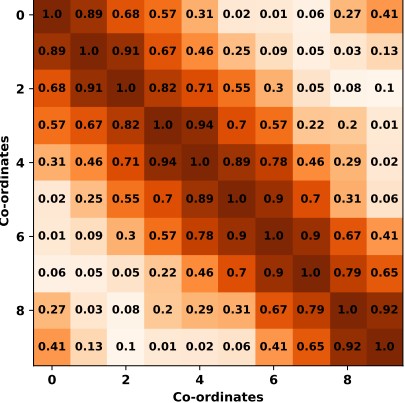

**Figure 2:** Cosine Similarity score for the learnt positional embeddings on the toy dataset by MET.

## 5 EXPERIMENTS

We now present an extensive empirical study of our proposed approach MET against various tabular-SSL methods, along with other classical supervised-learning baselines as described below. We experiment with standard tabular-SSL benchmarks namely permuted MNIST, permuted Fashion-MNIST and permuted CIFAR-10 that are used by existing works Verma et al. (2020); Ucar et al. (2021) as well as several real world datasets namely CovType, Adult-Income, Obesity, Arrhythmia,

**Table 1:** Downstream classification accuracy on four standard multi-class tabular datasets, comparing MET against various baselines. MET uses adversarial training + masking while MET-S only uses masking. MET either matches or outperforms the baselines across the all the datasets, giving an average accuracy improvement of 2% compared to the next best method of GBDT. Standard deviations are mentioned in appendix for completeness, although they are statistically insignificant.

| Type | Methods | FMNIST | CIFAR10 | MNIST | CovType | *Average* |
|---|---|---|---|---|---|---|
| Supervised Baseline | MLP | 87.62 | 16.50 | 96.95 | 65.47 | 66.64 |
| | RF | 88.43 | 42.73 | 97.62 | 71.37 | 75.04 |
| | GBDT | 88.71 | 45.7 | **100** | 72.96 | 76.84 |
| | RF-G | 89.84 | 29.32 | 97.65 | 71.57 | 72.10 |
| | MET-R | 88.84 | 28.94 | 97.44 | 69.68 | 71.23 |
| Self-Supervised Methods | VIME | 80.36 | 34.00 | 95.77 | 62.80 | 68.23 |
| | DACL+ | 81.40 | 39.70 | 91.40 | 64.23 | 69.18 |
| | SubTab | 87.59 | 39.34 | 98.31 | 42.36 | 66.90 |
| Our Method | MET-S | 90.94 | **48.00** | 99.01 | 74.11 | 78.02 |
| | MET | **91.68** | 47.82 | 99.19 | **76.71** | **78.85** |

Thyroid. Finally, we consider Criteo, which is an important large scale click-through rate prediction dataset with 45 million training samples. A detailed description of the datasets is in AppendixA.5.

## 5.1 BASELINES AND EXISTING METHODS

- VIME Yoon et al. (2020) : A SSL approach which uses a combination of masked token prediction and reconstruction loss. Note that MET proposes masked input reconstruction.

- SubTab Ucar et al. (2021) : SubTab views SSL as a multi-view representation learning problem, where representations from multiple croppings are aggregated at test time.

- DACL Verma et al. (2020) : A domain agnostic contrastive learning baseline which uses mixup as an augmentation. We specifically use DACL+ which uses geometric mean based mixup.

- MLP : We also compare against this natural baseline, wherein we train a MLP over the raw tabular data (and not the learnt representations) with the available labeled samples.

- Random Forest (RF): We train a random forest with 100 decision trees over the tabular data.

- Gradient Boosted Decision Trees (GBDT) Friedman (2001): GBDT is one of the most successful approaches on tabular data in general.

- Random Featurization (MET-R): We fine-tune an MLP over the representations from a randomly initialized encoder, checking the effectiveness of learnt representations in MET.

- Random Gaussian Featurization (RF-G) : Here, we compute random kitchen sink (Rahimi & Recht, 2008) style features, i.e., $\Phi(x) = Rx$ is the embedding of point $x \in \mathbb{R}^d$. Random features are known to be asymptotically an accurate approximation of the RBF kernel, which in turn is known to be a highly accurate and in fact, a "universal" classifier for tabular data. Note that we fix the embedding dimension of RF-G to be same as that of MET.

## 5.2 IMPLEMENTATION DETAILS

We use transformers Vaswani et al. (2017a) as the backbone for both the encoder and the decoder. Embedding dimension for the encoder and the decoder is chosen from a gridsearch in $\{64, 100, 128\}$, feedforward dimension from $\{64, 100, 128\}$, encoder and decoder depth from $\{1, 3, 6\}$ and the number of heads from $\{1, 2, 3\}$. For the tree based baselines (RF and GBDT), we do a gridsearch over the step size in $\{1.0, 0.3, 0.1, 0.01, 0.001, 0.0001\}$, maximum depth in $\{2, 5, 10, 20, \text{auto}\}$ and minimum samples in leaf node in $\{1, 2, 5\}$. The exact hyper-parameters for all the experiments are in appendix.

## 5.3 DOWNSTREAM CLASSIFICATION

In this section, we compare the quality of representations obtained from MET against various baselines as mentioned in Section 5.1. We compare the downstream classification accuracy using an

**Table 2:** Downstream accuracy and AUROC scores on five popular tabular datasets for binary classification, for MET and other baselines. MET outperforms all the baselines on both accuracy and AUROC metrics. Overall, MET achieves an average accuracy (AUROC) improvement of 5.40% (4.64%) over the next best method of GBDT. Standard deviations (statistically insignificant) are mentioned in appendix.

| Datasets | Metric | MLP | RF | GBDT | RF-G | MET-R | DACL+ | VIME | SubTab | MET |
|---|---|---|---|---|---|---|---|---|---|---|
| **Obesity** | Accuracy | 58.1 | 65.99 | 67.19 | 58.39 | 58.8 | 62.34 | 59.23 | 67.48 | **74.38** |
| | AUROC | 52.3 | 64.36 | 64.4 | 54.45 | 53.2 | 61.18 | 57.27 | 64.92 | **71.84** |
| **Income** | Accuracy | 84.36 | 85.88 | 86.01 | 85.59 | 75.51 | 84.46 | 82.23 | 84.43 | **86.25** |
| | AUROC | 89.39 | 91.53 | 92.5 | 90.09 | 83.48 | 89.01 | 87.37 | 88.95 | **93.85** |
| **Criteo** | Accuracy | 74.28 | 71.09 | 72.03 | 74.62 | 73.57 | 69.82 | 68.78 | 73.02 | **78.49** |
| | AUROC | 79.82 | 77.57 | 78.77 | 80.32 | 79.17 | 75.32 | 74.28 | 76.57 | **86.17** |
| **Arrhy-thmia** | Accuracy | 59.7 | 68.18 | 69.79 | 60.6 | 51.67 | 57.81 | 56.06 | 60.1 | **81.25** |
| | AUROC | 72.23 | 90.63 | 92.19 | 74.02 | 58.36 | 69.23 | 67.03 | 69.97 | **98.75** |
| **Thyroid** | Accuracy | 50 | 94.94 | 96.44 | 50 | 57.42 | 60.03 | 66.1 | 59.9 | **98.1** |
| | AUROC | 62.3 | 99.62 | 99.34 | 52.65 | 82.03 | 86.63 | 94.87 | 88.93 | **99.81** |
| *Average* | Accuracy | 65.29 | 77.22 | 78.30 | 65.84 | 63.39 | 66.89 | 66.48 | 68.99 | **83.69** |
| | AUROC | 71.21 | 84.74 | 85.44 | 70.31 | 71.25 | 76.27 | 76.16 | 77.87 | **90.08** |

MLP over the learnt representations in two cases : (a) when we use the entire labeled training data and (b) when we use only a fraction of the labeled training data.

**Multi-Class Classification Benchmarks :** Table 1 compares accuracy of MET with downstream classification against tabular-SSL methods and supervised learning baselines. Both MET (standard + adversarial reconstruction) and MET-S (only standard reconstruction) outperform the baselines across all the datasets. For example, on the permuted Fashion MNIST dataset, MET achieves an accuracy of 91.68%, outperforming all the other tabular SSL baselines like SubTab (87.59%) and also supervised baselines like GBDT (88.71%) and RF-G(89.84%). Similarly, on CovType, MET is about 10% more accurate DACL+, and in fact about 34% more accurate than SubTab, perhaps due to lack of semantics in neighbouring columns in the dataset. Here again, MET is around 4% better then the state-of-the-art classical baseline GBDT. Overall, MET establishes itself as a new state-of-the-art (SOTA) approach for self supervised learning on tabular data.

**Binary Classification Benchmarks :** Table 2 presents results on five real-world binary classification tabular datasets. MET again outperforms all other methods by a significant margin. In particular, on Criteo, which is a large-scale (45 million training sample) click prediction dataset, it achieves AUROC of 86.17% which is around 7.5% higher than that of GBDT and around 6% higher than the next best method RF-G. To put this in context, the SOTA AUROC on this dataset is $\approx 81\%$ (Wang et al., 2021) and has increased by less than 2% over the last six years. Overall, MET achieves an average AUROC (accuracy) improvement of 5.40% (4.64%) over the next best method of GBDT.

**Accuracy With A Fraction Of Labeled Training Data**: Next we do evaluation when only a fraction of labeled data is available for supervised training. Specifically, we vary the fraction of labeled data from 20% to 100% and compare the obtained downstream classification accuracies with the baselines. Figure 3 shows the variation of accuracy with the fraction of labeled data for MET, comparing against four of the strong baselines. We observe that MET outperforms the baselines for all the choices of fraction of labeled data used for supervised learning.

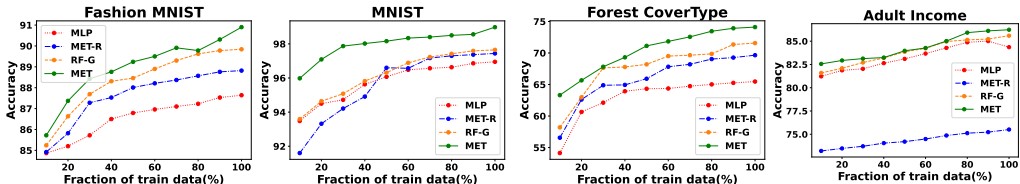

**Figure 3:** We compare the performance (downstream classification accuracy) of MET with various baselines as the fraction of labeled data used for training the downstream classifier is varied. Observe that MET consistently outperforms the baselines even when a smaller fraction of labeled data is used for downstream classifier training.

**Effect of Adversarial Reconstruction**: Next, we analyze the effect of using adversarial reconstruction loss, along with the standard reconstruction loss i.e., MET vs MET-S. In Table 1, we observe

that MET always outperforms or matches MET-S (without adversarial noise), and in some cases like CovType gives up to $2.5\%$ improvement compared to MET-S.

## 5.4 ANALYSIS AND ABLATION STUDIES

Here, we investigate whether MET is able to learn the underlying graphical model and also perform an ablation study to understand the effect of masking ratio and how the masked tokens are used.

**MET Learns The Underlying Graphical Model On CovType**: Similar to results in Section 4.3, we observe that the learned positional embeddings from MET indeed capture meaningful relations among the coordinates on real-world data too. For example, the position embedding of "Elevation" feature is maximally correlated (i.e., has highest inner product) with that of "Soil Type" feature. Indeed, several works in the forest science literature e.g., (Badía et al., 2016) have established the strong relation between elevation and soil type. MET is able to learn this relation through the masked reconstruction task. Similarly there is high correlation between the position embeddings of "Slope" and "Wilderness" features, and "Distance to roadways" and "Nearest wildfire ignition points" features, which again has been investigated in literature Mert & Yalcinkaya (2017); Francisco Moreira & Bacao (2010). More details on the features and correlations between their positional embeddings can be found in Appendix A.3.

Further, note from Table 1 and Table 2, MET is much more accurate than RF-G. While random features asymptotically approximate an RGF kernel itself, MET performs much better as it is able to capture the underlying graphical model using the reconstruction task.

**Masking Ratio** We observe that in general a high masking ratio ($\gamma$) of $50\% - 70\%$ seems to give the best downstream accuracy and hence good separable representations. Figure 4 in Appendix A.1.1 shows detailed plots of change in accuracy as masking ratio is varied across four datasets.

**Mask Token** Recall that MET uses a learnable mask token, which is passed to the decoder directly. Note that the mask token is kept same for all the masked coordinates (although positional encoding would be different) in MET. We tried a variant of MET, in which we used a separate mask token ($t_{\text{mask}}$) for each coordinate on the FMNIST and CovType datasets. In both cases, interestingly we observe that using a separate mask token didn't gave any performance gain. Similarly, passing the learnable masked token through the encoder didn't lead to any performance gain, despite the fact that this increases the compute requirement too.

Lastly, in the Appendix A.1, we study the effect of flattening (concatenating) the representations at inference time (Section 4.1) compared to averaging. We also show that downstream accuracy with MET representations is not sensitive to the depth of MLP over the representations.

## 6 CONCLUSION AND LIMITATIONS

In this paper, we proposed a *reconstruction based approach*, MET, for self-supervised representation learning on tabular datasets. MET is based on the hypothesis that there exists a latent (i.e., underlying/unobserved) graphical model which captures the relations between different coordinates of the tabular data. Conceptually, through experiments on an interesting toy setup, we show that MET is indeed able to learn the underlying graphical model, and hence gives much higher downstream classification accuracy. Practically, we show that over a suite of nine standard and real-world datasets, MET substantially outperforms a set of strong and diverse baselines, including other SOTA tabular-SSL methods as well SOTA supervised learning algorithms such as gradient boosted decision trees.

While reconstruction-based SSL has been shown to learn powerful representations across various domains such as text Devlin et al. (2019), vision He et al. (2021) as well as tabular (this paper), a thorough understanding of *why* reconstruction loss is capable of learning such good representations is still missing in literature. In this paper, we take a step in this direction by connecting masking-based reconstruction with the long line of work on learning graphical models. Our results motivate further investigation into the relation between graphical models and downstream classification tasks, as well as the efficacy of other graphical model learning approaches (e.g., those based on information theory) for self-supervised representation learning.

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

# A  APPENDIX

## A.1  ABLATION STUDY

In this section, we discuss various ablations and discuss the sensitivity of MET to various hyper-parameters.

### A.1.1  MASKING RATIO

We discussed the variation in performance of MET as we vary the fraction of input tokens masked ($\gamma$) in Section 5.4. Our general observation is that MET learns the best representations (in terms of downstream classification accuracy) when the masking ratio is high (50%-70%). Figure 4 shows the variation of downstream classification accuracy as the masking ratio is varied in $\{30, 50, 70, 80, 90\}$ on four different datasets. On FMNIST, MNIST and Adult-Income a masking ratio of 70% works the best whereas on CovType a masking ratio of 50% gives the best results.

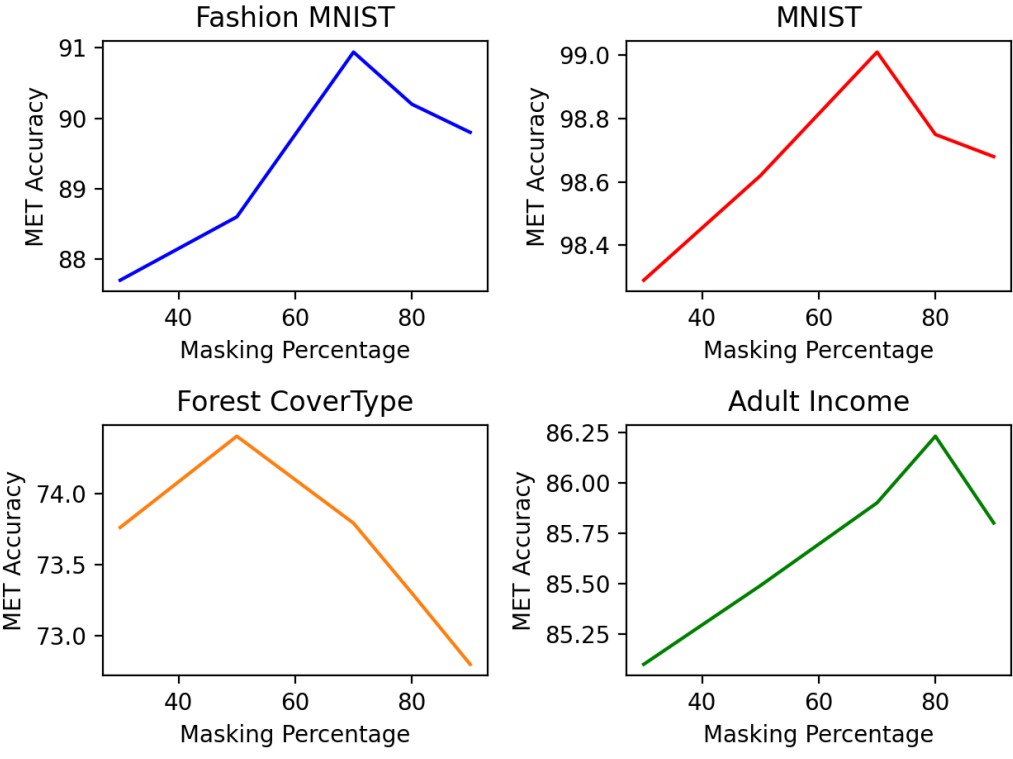

**Figure 4:** We study the variation in downstream accuracy as the masking ratio is varied in MET, for four tabular classification datasets. We observe that a high masking ratio (50%-70%) generally works the best.

### A.1.2  CONCATENATED VS AVERAGED EMBEDDINGS

In Section 4.1 we explained that at inference time, for an input $x$, the learnt representations are given by $\Phi(x) = T_{\theta_{enc}}(\psi(x^{[d]})) \in \mathbb{R}^m$ where $m = dd_{embed}$. Note that we made the choice of concatenating/flattening the learnt representations for each coordinate $j \in [d]$ in the encoder output. Here we perform an ablation by averaging the representations (instead of concatenation) across the coordinates to get the final learnt representation. We try both these approaches on FMNIST and CovType. For both datasets, concatenation significantly outperforms averaging. For CovType, concatenation and averaging obtain 76.71% and 61.87% accuracies respectively and for FMNIST, they obtain 91.68% and 88.64% respectively.

**Table 3:** Downstream task accuracy with varying number of hidden layers in the downstream MLP classifier with the learnt representation being fixed. We observe that for most datasets 2 or 3 hidden layers achieve the best performance, with relatively low sensitivity to the choice of the number of hidden layers.

| Num hidden layers | 0 | 1 | 2 | 3 | 4 | 5 |
|---|---|---|---|---|---|---|
| Accuracy | 87.16% | 90.59% | 90.86% | 90.82% | 90.65% | 90.62% |

### A.1.3 ACCURACY VS DEPTH OF MLP OVER LEARNT REPRESENTATIONS

Here we analyze how sensitive the downstream task performance is to variation in the depth of the MLP over the learnt representations. We fix the learnt representations for the FMNIST dataset and vary the number of hidden layers in the MLP. Table 3 shows the variation of accuracy with the hidden layers. We find that for most of the datasets, a depth of two to three hidden layers works the best. Although, on the same note, we mention that the downstream accuracy is not much sensitive to the choice of the number of hidden layers. For example, on FMNIST the best accuracy is 90.86% with 2 hidden layers whereas with 5 hidden layers, the accuracy is 90.62%.

### A.2 MEAN AND STANDARD DEVIATION FOR DOWNSTREAM CLASSIFICATION RESULTS

In Section 5.3, we compared the downstream classification accuracy for various baselines and MET over nine tabular datasets (Table 1, Table 2). For completeness, we add the results with mean and standard deviations, computed over five independent runs of each algorithm in Table 4 and Table 5.

**Table 4:** Downstream classification accuracies over four commonly used tabular benchmarks. MET uses adversarial training + masking for reconstruction-based self-supervised learning, whereas MET-S is an ablation where only masking is used. We observe that MET outperforms the baselines over all the datasets, establishing itself as a new state-of-the-art.

| Type | Methods | FMNIST | CIFAR10 | MNIST | CovType |
|---|---|---|---|---|---|
| Supervised Baselines | MLP | $87.57 \pm 0.13$ | $16.47 \pm 0.23$ | $96.98 \pm 0.1$ | $65.45 \pm 0.09$ |
| | RF | $87.19 \pm 0.09$ | $36.75 \pm 0.17$ | $97.62 \pm 0.18$ | $64.94 \pm 0.12$ |
| | GBDT | $88.71 \pm 0.07$ | $45.7 \pm 0.27$ | $100 \pm 0.0$ | $72.96 \pm 0.11$ |
| | RF-G | $89.84 \pm 0.08$ | $29.28 \pm 0.16$ | $97.63 \pm 0.03$ | $71.53 \pm 0.06$ |
| | MET-R | $88.81 \pm 0.12$ | $28.97 \pm 0.08$ | $97.43 \pm 0.02$ | $69.68 \pm 0.07$ |
| SSL Baselines | VIME | $80.36 \pm 0.02$ | $34.00 \pm 0.5$ | $95.74 \pm 0.03$ | $62.78 \pm 0.02$ |
| | DACL+ | $81.38 \pm 0.03$ | $39.7 \pm 0.06$ | $91.35 \pm 0.075$ | $64.17 \pm 0.12$ |
| | SubTab | $87.58 \pm 0.03$ | $39.32 \pm 0.04$ | $98.31 \pm 0.06$ | $42.36 \pm 0.03$ |
| MET (Our Method) | MET-S | $90.90 \pm 0.06$ | $\mathbf{47.96 \pm 0.1}$ | $98.98 \pm 0.05$ | $74.13 \pm 0.04$ |
| | MET | $\mathbf{91.68} \pm 0.08$ | $\mathbf{47.92 \pm 0.13}$ | $\mathbf{99.17 \pm 0.04}$ | $\mathbf{76.68 \pm 0.12}$ |

**Table 5:** Downstream accuracy and AUROC scores on five popular tabular datasets for binary classification, for MET and other baselines. MET outperforms all the baselines on both accuracy and AUROC metrics.

| Datasets | Metric | MLP | RF | GBDT | RF-G | MET-R | DACL+ | VIME | SubTab | MET |
|---|---|---|---|---|---|---|---|---|---|---|
| Obesity | Acc. | $58.1 \pm 0.07$ | $65.99 \pm 0.12$ | $67.19 \pm 0.04$ | $58.39 \pm 0.17$ | $58.8 \pm 0.59$ | $62.34 \pm 0.12$ | $59.23 \pm 0.17$ | $67.48 \pm 0.03$ | $\mathbf{74.38 \pm 0.13}$ |
| | auroc | $52.3 \pm 0.12$ | $64.36 \pm 0.07$ | $64.4 \pm 0.05$ | $54.45 \pm 0.08$ | $53.2 \pm 0.18$ | $61.18 \pm 0.07$ | $57.27 \pm 0.21$ | $64.92 \pm 0.06$ | $\mathbf{71.84 \pm 0.15}$ |
| Income | Acc. | $84.35 \pm 0.11$ | $84.6 \pm 0.2$ | $86.01 \pm 0.06$ | $85.57 \pm 0.13$ | $75.50 \pm 0.04$ | $85.99 \pm 0.24$ | $84.46 \pm 0.03$ | $84.41 \pm 0.06$ | $\mathbf{86.21 \pm 0.05}$ |
| | auroc | $89.39 \pm 0.2$ | $91.53 \pm 0.32$ | $92.5 \pm 0.08$ | $90.09 \pm 0.57$ | $83.48 \pm 0.23$ | $89.01 \pm 0.4$ | $87.37 \pm 0.07$ | $88.95 \pm 0.19$ | $\mathbf{93.85 \pm 0.33}$ |
| Criteo | Acc. | $74.28 \pm 0.32$ | $71.09 \pm 0.05$ | $72.03 \pm 0.03$ | $74.62 \pm 0.08$ | $73.57 \pm 0.12$ | $69.82 \pm 0.06$ | $68.78 \pm 0.13$ | $73.02 \pm 0.08$ | $\mathbf{78.49 \pm 0.05}$ |
| | auroc | $79.82 \pm 0.17$ | $77.57 \pm 0.1$ | $78.77 \pm 0.04$ | $80.32 \pm 0.16$ | $79.17 \pm 0.17$ | $75.32 \pm 0.27$ | $74.28 \pm 0.39$ | $76.57 \pm 0.05$ | $\mathbf{86.17 \pm 0.2}$ |
| Arrhy- thmia | Acc. | $59.7 \pm 0.02$ | $68.18 \pm 0.02$ | $69.79 \pm 0.12$ | $60.6 \pm 0.05$ | $51.67 \pm 0.1$ | $57.81 \pm 0.47$ | $56.06 \pm 0.04$ | $60.1 \pm 0.1$ | $\mathbf{81.25 \pm 0.12}$ |
| | auroc | $72.23 \pm 0.06$ | $90.63 \pm 0.08$ | $92.19 \pm 0.05$ | $74.02 \pm 0.12$ | $58.36 \pm 0.32$ | $69.23 \pm 0.98$ | $67.03 \pm 0.27$ | $69.97 \pm 0.07$ | $\mathbf{98.75 \pm 0.04}$ |
| Thyroid | Acc. | $50 \pm 0.0$ | $94.94 \pm 0.1$ | $96.44 \pm 0.07$ | $50 \pm 0.0$ | $57.42 \pm 0.37$ | $60.03 \pm 0.05$ | $66.1 \pm 0.19$ | $59.9 \pm 0.16$ | $\mathbf{98.1 \pm 0.08}$ |
| | auroc | $62.3 \pm 0.12$ | $99.62 \pm 0.03$ | $99.34 \pm 0.02$ | $52.65 \pm 0.13$ | $82.03 \pm 0.26$ | $86.63 \pm 0.1$ | $94.87 \pm 0.03$ | $88.93 \pm 0.12$ | $\mathbf{99.81 \pm 0.09}$ |

### A.3 CORRELATION ACROSS FEATURES FOR CovType

In Section 5.4, we discussed that MET is indeed able to capture meaningful relation among the coordinates on the CovType dataset. Figure 5 shows the cosine similarity of the learnt positional encoding of a feature/coordinate with the learnt positional encodings of other features/coordinates. Note that we show the average similarity across different runs of the MET on CovType, to reduce the variance. We observe that the feature "Elevation's" positional encodings are maximally correlated to

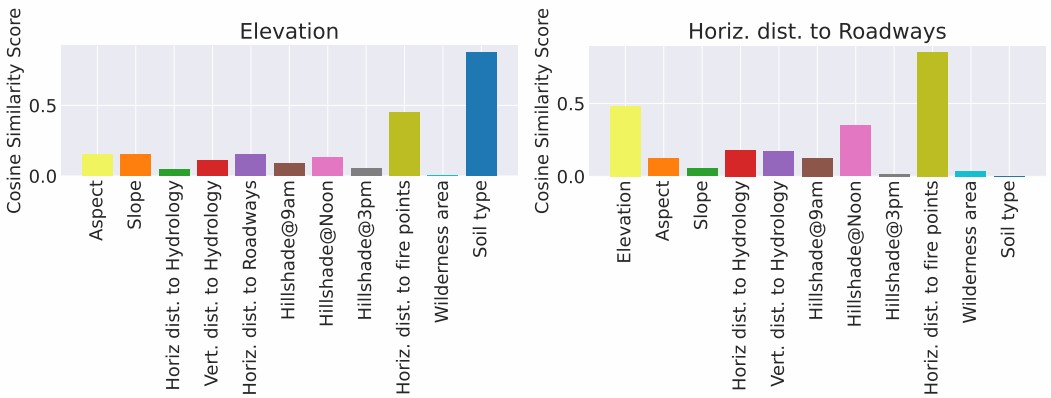

**Figure 5:** Comparing the cosine similarity of the learnt positional encoding of a feature/coordinate with the learnt positional encodings of other features/coordinates. The maximally correlated features are indeed meaningful.

the encodings of the feature "Soil Type", as has been corroborated in Badía et al. (2016) in the forest science literature. Similarly, we can observe that "Distance to roadways" has the most correlation with "Distance to nearest ignition point" as has been talked about in Francisco Moreira & Bacao (2010).

### A.4 RESULTS FOR TOY DATASET

In Section 4.3 we discussed results on a toy dataset from a linear graphical model. We observed that MET is able to capture the graphical model, as shown by the high correlation between the positional embeddings for the consecutive coordinates in Figure 2. Consequently, MET is able to perform the downstream task with high accuracy. For completeness, we share the accuracy and AUROC numbers for all the baselines in Table 6.

**Table 6:** We compare the performance of MET on the toy dataset from a linear graphical model with all the baselines. MET is able to capture the linear graphical model as shown in the correlation plot in Figure 2 and outperforms all the baselines both in accuracy and AUROC.

| Type | Methods | Accuracy | AUROC |
|---|---|---|---|
| | MLP | 68.65 | 75.4 |
| | RF | 82.25 | 90.16 |
| Supervised Baseline | GBDT | 88.4 | 95.35 |
| | RF-G | 66.6 | 73.62 |
| | MET-R | 73.48 | 80.29 |
| | DACL+ | 74.65 | 82.92 |
| Self-Supervised Methods | VIME | 62.65 | 71.19 |
| | SubTab | 63.28 | 73.52 |
| Our Method | MET | **96.7** | **99.61** |

### A.5 DATASETS

**MNIST**[1]: It is a 10-class classification handwritten digit recognition dataset. Input images of size $28 \times 28$ are flattened to get a 784-dimensional vector as the input. The flattened dataset is then permuted (i.e. the pixel/coordinate order is shuffled) to get the final dataset. A split of $54,000$ samples as train set, 6000 samples as validation set and $10,000$ samples as test set is used.

**FMNIST**: Fashion-MNIST(*FMNIST*) consists of $28 \times 28$ dimensional images, and is a 10-class classification dataset. The data is flattened to get 784 coordinates in tabular form and then shuffled

---

[1]The data is normalized and shuffled for all datasets.

as mentioned above. A split of $54,000$ samples as train set, 6000 samples as validation set and $10,000$ samples as test set is used.

**CIFAR-10**: It is a colored dataset with images of size $32 \times 32 \times 3$ belonging to ten different classes. We flatten it to get 3072 coordinates in tabular form and then shuffle as mentioned previously. A split of $45,000$ samples as train set, $5000$ samples as validation set and $10,000$ samples as test set is used.

**Forest Cover Type**: The covtype dataset is a 7 class classification UCI dataset where the task is to predict the forest cover type. The dataset has in total 54 features, out of which there is a one hot vector of length 4 representing the wilderness. We replace this one hot vector with a single integer denoting the index of non-zero entry. Similarly there is a one hot vector of length 40 for soil type, which we again replace with a single integer denoting the index of the non-zero entry. In summary, we process the 54-dimensional dataset to a much harder 12-dimensional dataset since the categorical features are now represented as integers instead of one-hot representations[2]. A split of $11,340$ samples as train set, 3780 samples as validation set and $565,892$ samples as test set is used.

**Income**: The adult income dataset is a UCI dataset where the prediction task is to determine whether a person makes over \$50K a year based on the census data. It consists of a mix of six continuous and eight categorical fields. Similar to CoverType dataset, we use integers instead of one-hot representation for the categorical features and get 14 features in a tabular form. A split of $27,146$ samples as train set, 3016 samples as validation set and $15,060$ samples as test set is used.

**Obesity**: The obesity dataset has a task of predicting obese vs non-obese based on the human gut metagenomic sample. The input has 465 features. The features are normalized and shuffled before feeding it to the model. It is a relatively small dataset of 253 samples, hence we use a 80-10-10 train-val-test split with 10-fold cross-validation for evaluation purposes.

**Criteo**: Display advertising is a billion-dollar industry and an important use case of machine learning. Criteo consists of one-week data from CriteoLabs for click-through-rate(*CTR*) prediction summing up to 45M samples with 39 features each. Out of the 39 features, 26 are categorical, some of which have as many as 5M distinct values in the form of anonymized string and the other 13 are real-valued fields. A split of $34,380,462$ samples as train set, $3,438,047$ samples as validation set and $9,168,124$ samples as test set is used.

**Arrhythmia**: It is a standard outlier detection (one class classification) benchmark for tabular dataset, available on OODS-arrhythmia repository, which we convert into a binary classification one. The dataset has a set of inlier samples and a set of outlier samples which are much less in number compared to the inlier samples. We split the outlier samples equally between the train and the test set. We keep a $1:1$ ratio between the inlier vs outlier classes in the test set and use the remaining inlier samples in the training set. A split of $348$ samples as train set, 38 samples as validation set and 66 samples as test set is used.

**Thyroid**: Again, it is a standard outlier detection benchmark for tabular dataset, which we convert into a binary classification one. It is available on the thyroid dataset repository. We follow the same procedure as the Arrhythmia dataset above for splitting the inlier and outlier samples into train-test set. A split of 6000 samples as train set, 666 samples as validation set and $534$ samples as test set is used.

## A.6 HYPER-PARAMETERS

In this section, we share the hyper-parameters of MET for replicating the results on all the nine tabular datasets. Note that Encoder Depth refers to the number of transformer layers in the encoder and Decoder Depth refers to the number of transformer layers in the decoder. Adversarial

---

[2]Consequently, our accuracy numbers are not directly comparable to standard results on this dataset.

Learning Rate ($lr_{adv}$) refers to the step size for gradient ascent in adversarial training and lr refers to the learning rate for gradient descent on reconstruction loss. We perform a grid-search for embedding dimension($d_{embed}$) and feed-forward dimension(fw) in $\{64, 100, 128\}$, number of heads in the transformer architecture in $\{1, 2, 3\}$, encoder and decoder depth in $\{1, 3, 6\}$, learning rate($lr$) in $\{1e^{-1}, 1e^{-2}, 1e^{-3}, 1e^{-4}, 1e^{-5}\}$, masking ratio($\gamma$) in $\{30, 50, 70, 80, 90\}$, adversarial steps($adv_{steps}$) in $1, 2, 4$, radius of L2-norm ball($\epsilon$) in $\{2, 6, 10, 12, 14\}$ and step size for gradient ascent in ($lr_{adv}$) in $\{0.1, 0.01\}$. The optimal set of hyper-parameters for various datasets obtained are mentioned in Tables7 and 8. Note that all the ablation studies are conducted using these parameters.

**Table 7:** Hyper-parameters for replicating the results with MET.

|  | Embedding Dimension($d_{embed}$) | Feed-forward Dimension | Number of Heads | Encoder Depth | Decoder Depth | lr | Masking Ratio($\gamma$) |
|---|---|---|---|---|---|---|---|
| FMNIST | 64 | 64 | 1 | 6 | 1 | $1e^{-5}$ | 0.70 |
| CIFAR10 | 100 | 64 | 2 | 3 | 3 | $1e^{-4}$ | 0.70 |
| MNIST | 64 | 64 | 1 | 6 | 1 | $1e^{-4}$ | 0.70 |
| CovType | 100 | 64 | 1 | 1 | 1 | $1e^{-4}$ | 0.50 |
| Obesity | 64 | 100 | 3 | 6 | 3 | $1e^{-5}$ | 0.70 |
| Income | 64 | 64 | 1 | 3 | 6 | $1e^{-3}$ | 0.80 |
| Criteo | 64 | 100 | 1 | 6 | 6 | $1e^{-4}$ | 0.50 |
| Arrhythmia | 64 | 64 | 1 | 1 | 1 | $1e^{-3}$ | 0.30 |
| Thyroid | 100 | 128 | 2 | 3 | 6 | $1e^{-3}$ | 0.80 |

**Table 8:** Hyper-parameters for replicating the results with MET.

|  | Adversarial Steps($adv_{steps}$) | L2 Norm Ball Radius($\epsilon$) | Adversarial $lr$ ($lr_{adv}$) |
|---|---|---|---|
| FMNIST | 2 | 2 | $1e^{-2}$ |
| CIFAR10 | 3 | 14 | $1e^{-2}$ |
| MNIST | 2 | 12 | $1e^{-2}$ |
| CovType | 5 | 4 | $1e^{-1}$ |
| Obesity | 5 | 3 | $1e^{-3}$ |
| Income | 1 | 6 | $1e^{-1}$ |
| Arrhythmia | 1 | 12 | $1e^{-2}$ |
| Thyroid | 2 | 2 | $1e^{-3}$ |

## A.7 COSINE-SIMILARITY AND LATENT GRAPH

In this section, we show how cosine-similarity is co-related with the ability to capture the latent graph structure. The experimental setup tries to reconstruct a given coordinate only by using another coordinates. Given a set of coordinates $x_i$'s we fix a coordinate $t = x_i$ that needs to be reconstructed. Now by using the other set of coordinates $x_{j!=i}$ one at a time, we reconstruct $t$ and measure the mean-absolute error across all samples of the toy dataset4.3. As shown in Figures 6, 7 and 8 we observe that features with high cosine similarity scores show lower mean absolute error while reconstruction. Note that we min-max normalize the cosine similarity score and the mean absolute errors for plotting them on the same scale.

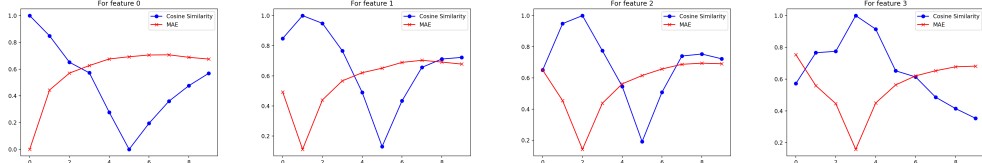

**Figure 6:** Higher cosine similarity score leads to better reconstruction of features and hence is able to capture the latent graph structure. The following graphs are for coordinates 1,2,3,4.

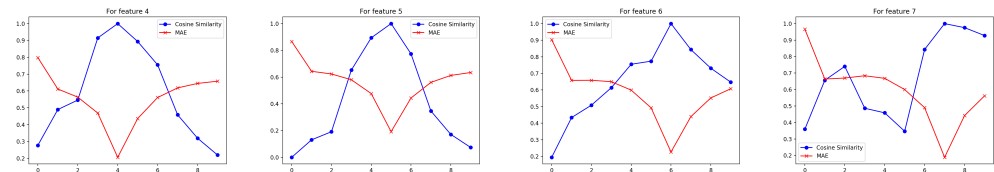

**Figure 7:** Higher cosine similarity score leads to better reconstruction of features and hence is able to capture the latent graph structure. The following graphs are for coordinates 5,6,7,8.

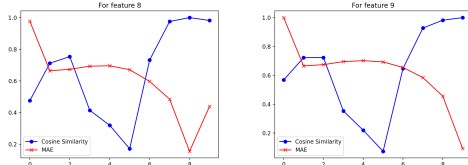

**Figure 8:** Higher cosine similarity score leads to better reconstruction of features and hence is able to capture the latent graph structure. The following graphs are for coordinates 9,10.

