# OpenReview forum: "MET : Masked Encoding for Tabular data"
_ICLR.cc/2023/Conference — Submitted to ICLR 2023_

### Official Review · Reviewer_twHD · 2022-10-21

**Confidence:** 3
**Correctness:** 3
**Technical Novelty And Significance:** 2
**Empirical Novelty And Significance:** 3
**Recommendation:** 6

**Clarity, Quality, Novelty And Reproducibility:**

The detailed comments are described in the previous section. In summary,
- Clarity :: This paper is well-written, and it is easy to follow.
- Quality :: The empirical results are strong.
- Novelty :: I cannot find methodological novelty in this paper.
- Reproducibility :: All the hyperparameters are well-described.


**Strength And Weaknesses:**

Strengths
- The proposed method, MET, outperforms recent SSL methods (VIME, DACL, SubTab) and classical ML approaches (RF, GBDT, RF-G) by a large margin.
- This paper empirically verifies that the learned positional embeddings well capture the relationship between features; in other words, MET is highly interpretable.

Weaknesses
- I feel the lack of methodological novelty in this paper. I think this paper is a simple application of the existing MAE approach into the tabular domain with no modification.
- Most multi-classification benchmarks are vision datasets, not tabular: FMNIST, CIFAR10, and MNIST. Other tabular benchmarks should be tested rather than vision datasets.
- How to select the hyperparameters in Table 7? Which split (train, validation, test) is used for the selection?
- How to handle categorical features? This should be described in the main manuscript.
  - I found that this paper uses categorical features as integers instead of one-hot vectors for CovType. Why use this approach?
- The transformer architecture is necessary? Since the number of input features is constant, one can use MLPs (or other architectures) as VIME did. I cannot find why the transformer architecture is important.


**Summary Of The Paper:**

This paper proposes a self-supervised learning (SSL) framework for tabular data, Masked Encoding for Tabular Data (MET). MET employs masked auto-encoding with Transformer architectures. This paper shows that MET outperforms existing SSL methods for tabular data and classical machine learning approaches. In addition, this paper empirically verifies that the learned representations well capture the relationship between features.


**Summary Of The Review:**

Although the proposed method is not new, the empirical results are strong. Since the tabular domain is yet under-explored, I think this paper's contribution is somewhat meaningful in this field. Hence, I vote for weak accept.

---

> ### Author Response · Authors · 2022-11-15
> **Response to clarifications requested by reviewer twHD**
>
> We thank the reviewer for their time, and recognizing the “strong empirical effectiveness” and interpretability of the method. Below we try to address some of the key concerns raised.
>
> $\textbf{Novelty}$ : Our work has two key contributions :
> * We are the first to explore the use of masked reconstruction based approaches at a coordinate level for self-supervised learning in domain agnostic settings of tabular data to the best of our knowledge. Further, although masked reconstruction based approaches have been successfully explored in NLP [1] and computer vision [2], our paper is the first to connect this to the long line of work on graphical model learning, by showing through toy experiments that this structure is indeed captured by the “learnt” positional encodings. We extended similar analysis to real world dataset as well in Section 5.4
> * More importantly, we propose to search for adversarial perturbations in the input space, to find data points (close to the training data) in the input manifold which are specifically hard to reconstruct and hence hinder good representation learning.
>
> Combined, our proposed approach MET gives consistent and substantial accuracy (and ROC) improvements across a range of standard as well as real world tabular datasets from high impact settings (including SOTA results on large scale datasets with 45 million samples). Hence MET makes a strong contribution to the area of self-supervised learning in domain agnostic settings.
>
> $\textbf{Dataset}$: Note that all our image datasets were $\textbf{permuted along the feature dimension, to remove any kind of “spatial” structure}$. Our motivation to experiment (also) on these “permuted” MNIST, “permuted” CIFAR and “permuted” FMNIST was to benchmark with popular “deep learning” based approaches in the recent literature like VIME[4] and SubTab[5].
>
> $\textbf{Train-Val split}$ : All our hyper-parameters are tuned on the validation sets. For the datasets where a publicly available validation set is missing, we split the training dataset in 90:10 ratio to create a training-validation set. The same has been described in detail for each dataset in Appendix A5.
>
> $\textbf{Categorical Variables}$ : We use integers for categorical features : (a) To reduce the dimensionality of the data. (b) Trying to reconstruct a long one-hot vector can unnecessarily increase the complexity of the task, and make it hard to learn the latent graphical model.
>
> $\textbf{Transformer Architecture}$ : Our proposed approach MET is based on learning the underlying graphical structure of the tabular data. We aimed to capture this via the attention mechanism of Transformers. For example, see the toy experiment in Section 4.3, where we show that the relation between the consecutive coordinates of the toy dataset is indeed captured by the “learnt” positional encodings in the transformer. Learning this structure in the pretraining dataset is quite crucial to generalize to a variety of downstream classification tasks.
>
> Further, we would like to point the reviewer to the discussion in Section 5.4, where we show that even in real-world datasets, our proposed approach MET is able to capture meaningful correlations among the features. For example, on CovType, MET has maximally correlated positional embeddings for “Soil Type” and “Elevation” features of the dataset, which indeed is correct as validated in forest science literature [3].
>
> References.
> [1] Jacob Devlin, Ming-Wei Chang, Kenton Lee, and Kristina Toutanova. BERT: pre-training of deep bidirectional transformers for language understanding. 2018.
> [2] Kaiming He, Xinlei Chen, Saining Xie, Yanghao Li, Piotr Dollár, and Ross B. Girshick. Masked autoencoders are scalable vision learners, 2021.
> [3] David Badıa, Alberto Ruiz, Antonio Girona, Clara Mart´ı, Jose Casanova, Paloma Ibarra, and Raquel ´ Zufiaurre. The influence of elevation on soil properties and forest litter in the siliceous moncayo massif, sw europe. Journal of Mountain Science, 13(12):2155–2169, 2016.
> [4] Jinsung Yoon, Yao Zhang, James Jordon, and Mihaela van der Schaar. Vime: Extending the success of self-and semi-supervised learning to tabular domain. Advances in Neural Information Processing Systems, 33:11033–11043, 2020.
> [5] Talip Ucar, Ehsan Hajiramezanali, and Lindsay Edwards. Subtab: Subsetting features of tabular data for self-supervised representation learning. Advances in Neural Information Processing Systems, 34, 2021.

---

> > ### Comment · Reviewer_twHD · 2022-12-01
> > **Response to Authors**
> >
> > Thank you for your efforts and time in your response.
> >
> > **[Novelty]** I agree that this work is the first to incorporate Masked Auto Encoding (MAE) into tabular data. I would like to give credit for your contribution in tabular data. However, adversarial reconstruction SSL was already studied in language, e.g., COCO-LM [1]. So I'm still concern about the limited technical novelty (as also mentioned by Reviewer F52Q). So I would like to keep my score as Week Accept.
> >
> > **[Image Benchmarks]** Thank you for the explanation. I agree that the datasets are similar to tabular benchmarks due to the permutation. IMHO, the tabular learning literature (not only your work) should consider more real tabular benchmarks instead of synthetic ones.
> >
> > **[Hyperparameter search]** Thank you for the explanation.
> >
> > **[Categorical Variables]** I still feel that this is somewhat awkward because categorical variables should be differently treated compared to continuous variables.
> >
> > **[Transformer]** Thank you for the explanation. As you said, I also think identifying correlations between columns is a meaningful task for tabular data. That could be a merit of the architecture.
> >
> > [1] Meng et al., COCO-LM: Correcting and Contrasting Text Sequences for Language Model Pretraining, NeurIPS 2021

---

> > > ### Author Response · Authors · 2022-12-05
> > > **Clarification about differences with COCO-LM and Novelty**
> > >
> > > We thank the reviewer for their time to go through our response.
> > >
> > > $\textbf{Novelty:}$ Adversarial perturbations in MET are significantly different from COCO-LM. COCO-LM tries to learn tokens that are more deceiving than mask tokens for the masked coordinates. However, in MET, we perturb the whole input to find points in the input manifold where the reconstruction loss is high. These "adversarial" points are then masked, appending the same masked token as the one used for non-adversarial points, and passed through the encoder and decoder to get the reconstruction loss.
> > > In essence, MET tries to learn a more consistent classifier in the input space, $\textbf{significantly different}$ from COCO-LM, where the perturbations of the masked token are done for regularization and stability of the training.
> > >
> > > Coming up with a more nuanced way to handle categorical variables would be an interesting direction for future work. We believe that our work takes an important initial step in the direction of modeling the tabular learning problem through the lens of a graphical model learning.
> > >
> > > We thank the reviewer for agreeing that "identifying correlations between the columns is a meaningful task", and will be more than happy to engage in further discussion to assist in a more positive evaluation of the work.

---

### Official Review · Reviewer_wvLH · 2022-10-25

**Confidence:** 4
**Correctness:** 3
**Technical Novelty And Significance:** 2
**Empirical Novelty And Significance:** 2
**Recommendation:** 3

**Clarity, Quality, Novelty And Reproducibility:**

The paper is reasonably clear to read. I had no difficulties reading it.

I do not believe that I have already seen the specific combination of adversarial masked loss and transformers.

The accompanying code will help in terms of reproducibility. However, it is not sufficient, as I wasn't able to understand the whole hyper-parameter selection / training routines.

**Strength And Weaknesses:**

Transformers are clearly powerful architectures and it is interesting to mix them with masked loss for tabular data.

The datasets benchmarked in table 1 are not tabular data (4 out of 5 are image data). Grinsztajn et al "Why do tree-based models still outperform deep learning on typical tabular data?" give a clear definition of properties of tabular data and a list of openly-available data to use.

What is the train-test-validation strategy used? Reading the code submitted withthe manuscript seems to reveal that the left-out data set where the performance of the algorithm is evaluated was also used to set the hyper-parameters.

How are categorical columns dealt with in the reconstruction loss?

How are categorical inputs dealt with in the perturbation of the input in eq 6?



**Summary Of The Paper:**

This submission contributes a masking based reconstruction approach with a transformer architecture for self-supervised learning based on quadratic reconstruction error with adverserial perturbation of the input.

The method is benchmarked on a set of classic datasets where it is shown to predict better than standard methods including gradient boosted trees.

**Summary Of The Review:**

This manuscript contributes an interesting idea, but it is not clear that the empirical evidence is entirely robust.

---

> ### Author Response · Authors · 2022-11-15
> **Response to clarification requested by reviewer wvLH**
>
> We thank the reviewer for their time and recognizing our work as an “interesting idea”.  We evaluated the effectiveness of our proposed approach of a suite of 9 benchmarks, out of which $\textbf{6 are non-image datasets}$, including a large scale real-world benchmark of Criteo with $\textbf{45 million training}$ samples. Hence, we politely disagree with the reviewer that our evaluation is not “robust”. Below we try to address some of the concerns raised in detail.
>
> $\textbf{Datasets}$: First, we would like to highlight that $\textbf{only 3/9}$ of the datasets we evaluate on are “permuted” image data. However, note that all our image datasets were $\textbf{permuted along the feature dimension, to remove any kind of “spatial” structure}$. Our motivation to experiment (also) on these “permuted” MNIST, “permuted” CIFAR and “permuted” FMNIST was to benchmark with popular “deep learning” based approaches in the recent literature like VIME[1] and SubTab[2].
>
> That said, we would like to highlight that $\textbf{6/9}$ of our datasets are not from “permuted image” domain, and our proposed approach MET actually gives much higher gains on these datasets. For example, on CovType, our approach outperforms the most competitive baseline of gradient boosting by $3.75\%$. On the real world click prediction dataset of “Criteo”, MET outperforms the current SOTA [3] by $5\%$. To put the numbers in perspective, we would also like to note that the state-of-the-art of this dataset has increased by less than $2\%$ in the past 2 years.
>
> We thank the reviewer for pointing us to “Grinsztajn et al”. However we do not agree that having all the features from the same domain necessarily excludes the data from being “tabular”. Whenever a data lacks a “structure” or “ordering” (for example, semantic or temporal) within it’s features, it should be classified as a tabular data.
>
> $\textbf{Categorical Variables}$ : We chose to convert the categorical one-hot labels to their corresponding integer labels and then consider them simply as regression variables, as described in Appendix A5 for the CovType and AdultIncome datasets. Dealing with adversarial perturbations of these variables follows.
>
> $\textbf{Code issue}$ : We apologize for submitting a redundant version of the code. We have re-submitted the correct version of our code, which the reviewer can run directly out-of-the-box and verify. The code outputs accuracies on the validation set for various hyper-parameters, and the test accuracy corresponding to the best hyper-parameter based on the validation accuracy matches what is reported in the original paper. For the datasets where publicly available validation is missing, we split the training dataset in 90:10 ratio to create a training-validation set.
>
> References.
> [1] Jinsung Yoon, Yao Zhang, James Jordon, and Mihaela van der Schaar. Vime: Extending the success of self-and semi-supervised learning to tabular domain. Advances in Neural Information Processing Systems, 33:11033–11043, 2020.
> [2] Talip Ucar, Ehsan Hajiramezanali, and Lindsay Edwards. Subtab: Subsetting features of tabular data for self-supervised representation learning. Advances in Neural Information Processing Systems, 34, 2021.
> [3] Zhiqiang Wang, Qingyun She, and Junlin Zhang. Masknet: Introducing feature-wise multiplication to ctr ranking models by instance-guided mask. ArXiv, abs/2102.07619, 2021.

---

> > ### Comment · Reviewer_wvLH · 2022-11-27
> > **Still worried about the representativeness of the benchmarks**
> >
> > I thank the authors for their answer. Having read it and given it a lot of thoughts, I still worry about the representativeness of the benchmarks. When I work on a tabular dataset, it has columns containing things like "age", "sex", or "temperature", "pressure". Shuffling these values across these columns does not make any sense. It is a dataset of a different nature than a shuffled image data.
> >
> > I do not find that the current study shows evidence that the model actually works well on tabular data.

---

> > > ### Author Response · Authors · 2022-11-28
> > > **Clarifying the permutation of image datasets**
> > >
> > > We thank the reviewer for their response.
> > >
> > > We would like to clarify again that the permutation is same for every row in the image dataset, i.e. we permute all the images with the same permutation along the feature dimension. This is done simply to ensure that their is no spatial structure which can be used for classification.
> > > Again, note that this permutation is done only for the image datasets. For datasets like CovType, Arrythmia, Criteo, etc., as considered in Table 2, we do not permute since as the reviewer pointed out,  table remains the same if we have data as {"age", "gender" , "location"} or {"location", "age" , "gender"}. Hence, their is no inconsistency in evaluation.
> > >
> > > Further, we would request the reviewer to also look at 5 other datasets in Table 2. We iterate again, that our work considers image datasets only to compare against previous benchmarks.

---

> > > > ### Comment · Reviewer_wvLH · 2022-12-14
> > > > **Understood, thank you for clarifying**
> > > >
> > > > I understand what you are doing (I was confused for a while).
> > > >
> > > > However, this column switching will slightly obfuscate the structure but in no way make it disappear. It is still the case that taking the mean of col i with eg col j is equivalent to taking the mean of two neighboring voxels (i and j have to be learned), which is a natural interpolation of the image value in between those two voxels. In no way does that correspond to say age and weight, where taking the mean is not a natural quantity (in particular given that these two values are in different units).
> > > >
> > > > Once again I would like to stress: these are not the typical columnar data that we find in applications, eg healthcare, add placement, educational data mining, sales.

---

### Official Review · Reviewer_F52Q · 2022-11-15

**Confidence:** 3
**Correctness:** 3
**Technical Novelty And Significance:** 2
**Empirical Novelty And Significance:** 3
**Recommendation:** 5

**Clarity, Quality, Novelty And Reproducibility:**

- Clarity: The paper is overall good to follow. However, the authors emphasize the connection to graphical models quite frequently and this aspect is unclear to me: GMs provide a convenient way to encode joint probability distributions (e.g., their conditional independence structures). Similarity of positional encodings seems a relatively weak analogy in this context. If the connection to GMs is to be used as a motivation for this approach (as it is currently in the paper), it would like to see a stronger theoretical justification and/or empirical evaluation.
- Reproducibility: The large improvements of MET over even supervised baselines on some datasets seem surprising. For instance, on Arrythmia, GBDTs achieve ~70% accuracy and all other baselines, including SSL methods for tabular data are (well) below these results. Yet, MET seems to achieve 81+% accuracy. Can the authors provide more context on why this is the case? The very low results of 50% acc for baselines MLPs and RF-G on Thyroid (e.g., random results on binary classes) would also benefit from more context.
- Reproducibility: The paper overlaps only on a few datasets with prior work (e.g., VIME, Subtab). However, on these datasets, the results do not seem to match those reported in prior work. For instance, for VIME on Income, the original paper reports ~88% accuracy, what would surpass the results of MET. Is this due to different evaluation setups? If yes, what is the motivation for using a different setup?
- Reproducibility: Related to my question above: while the authors report hyperparameter ranges (which is great) the exact training setup (e.g., splits, hyperparameter selection, stopping criteria) is unclear
- Novelty: Methodologically, the approach is relatively straightforward (random masking + concatenation of positional embeddings). In terms of technical contributions, I found novelty therefore to be somewhat limited. However, if my questions with regard to clarity and reproducibility above would be addressed, this would be less of a concern as the method shows promising results.
- Minor: Evaluation on image datasets is somewhat confusing. As the authors note themselves, images have very different properties than tabular data (e.g., non-exchangeability).

**Strength And Weaknesses:**

The paper addresses an interesting problem, i.e., how to extend the success of self-supervised representation learning to tabular data. The proposed approach is reasonable and seems to provide promising experimental gains. The experimental evaluation is performed on a wide range of datasets what surpasses prior work in this area. Due to the use of learnable positional embeddings, the methods gains also some form of interpretability (i.e., similarity relations between features) what is a nice additional benefit of the approach.

However, in its current form, the paper has some shortcomings that would require further attention, most importantly with regard to clarity, novelty, and reproducibility. I will detail these aspects further in the next section.

**Summary Of The Paper:**

The paper proposes a new model for self-supervised learning on tabular data which is based on transformers and auto-encoding via masked reconstruction. To encode feature types, the model uses learnable "positional" embeddings which can also capture relations between features. The method is evaluated on a variety of tabular and non-tabular datasets where it shows promising results.

**Summary Of The Review:**

The paper explores interesting ideas for SSL on tabular data and shows promising results. However, short-comings in terms of clarity, reproducibility would require attention prior to publication.

---

> ### Author Response · Authors · 2022-11-19
> **Response to Clarifications requested by Reviewer F52Q**
>
> We thank the reviewer for their time and recognizing the strong effectiveness of our proposed approach across a “ wide range of datasets what surpasses prior work in this area”. Below we try to address some of the concerns raised in detail.
>
> In section A7, we show how cosine-similarity is co-related with the ability to capture the latent graphical structure. Specifically, given a $10$-dimensional toy dataset similar to the experimental setting in Section 4.3, we use MET to reconstruct each coordinate using every other coordinate (and masking the rest) in the toy dataset. Our observations reveal that 2 co-ordinates with high cosine similarity of positional embeddings give lower reconstruction error as shown in Figures 6,7,8. This empirically highlights that features with similar positional embeddings are related in terms of their graphical strcuture too.
>
> $\textbf{Clarity}$: We add an experimental setup in section A.7 of appendix where we show how the trends across cosine similarity scores across features shows the capacity to capture the underlying latent graphical model. We also acknowledge that capturing the latent graphical model is just an intuition behind the developed method MET. In the experimental setup, we try to reconstruct a given co-ordinate by using the other co-ordinates of the toy dataset (one at a time) and are able to show that features with higher cosine similarity scores reconstruct each other easily. Hence this can help show that the latent graph structure at some level is being captured in the underlying representations.
>
> $\textbf{Reproducibility}$: We have shared the lists of exact hyper-parameters and code to reproduce the results as shared in the paper. We used the official codes provided by the authors for VIME and SubTab, however, we could not reproduce the reported results on Income dataset when the given grid-search in the respective papers is performed over the hyper-parameters. Although we were able to reproduce reported results on other overlapping dataset of permuted MNIST.
>
> We believe that large surprising gains over the baselines (GBDT as well as MLP and RF-G) is due to two factors : (a) Use of large architectures like transformer and more importantly (b) Learnable positional encodings learn important correlations between the various features of the tabular data, which helps MET to perform well on the downstream tasks.
>
> All our hyper-parameters are tuned on the validation sets. For the datasets where a publicly available validation set is missing, we split the training dataset in 90:10 ratio to create a training-validation set. The same has been described in detail for each dataset in Appendix A5. We early stop based on the ID Validation Accuracy.
>
> $\textbf{Datasets}$: Note that all our image datasets were $\textbf{permuted along the feature dimension, to remove any kind of “spatial” structure}$. Also, note that only $\textbf{only 3/9}$ of the datasets we evaluate on are “permuted” image datasets. Our motivation for choosing these datasets was to benchmark on more common/overlapping datasets with prior literature like VIME and SubTab.
>
> We hope we are able to address the concerns of the reviewer using the added experiment and more details around reproducibility.

---

> > ### Author Response · Authors · 2022-12-05
> > **Thanks to the reviewer**
> >
> > We again thank the reviewers for their feedback on our work. We hope that our response addresses the major concerns and would help with a more positive evaluation of the work. We are happy to engage in further discussions if needed.

---

### Decision · Program_Chairs · 2023-01-20

**Decision:**

Reject

**Justification For Why Not Higher Score:**

Too many unresolved concerns that make the results unreliable.

**Justification For Why Not Lower Score:**

Promising.

**Metareview: Summary, Strengths And Weaknesses:**

Two reviewers recommend rejection, and one recommends acceptance, but all reviews agree more or less about the strengths and weaknesses of the submission. Overall, I recommend rejection due primarily to concerns about the reliability of the experimental results.

The submission is not anonymous because the paper is uploaded at https://arxiv.org/abs/2206.08564.

Because only two reviews arrived on time, I wrote the following review myself, independently:


Review of MET: MASKED ENCODING FOR TABULAR DATA

Overall, this is a promising paper. The method proposed makes sense, and the results on the Criteo dataset are impressive. However, the design and results of other experiments are questionable, and the paper itself needs improvement.

My most important criticisms of the paper are the following:
1. Categorical features are represented as integers from 1 to the number of classes. This is a terrible input representation for most ML methods, so it can be a major handicap for competitor methods, making the comparison with them not fair. This is especially important for Criteo, where some categorical features are strings with literally millions of alternative values. See "On Embeddings for Numerical Features in Tabular Deep Learning" by Gorishniy et al.
2. The paper, and the appendix, and the code in the zip file, all fail to specify which modern GBDT method is used. This needs to be a method that handles categorical features in a non-trivial, state-of-the-art way. Moreover, the number of trees is not stated for GBDT, and 100 may be too few. The references for GBDTs are too old, from 1999 and 2001. The zip file does not contain the code that runs competitor methods, so the results in the submission are not reproducible, and we still do not know which GBDT method was used.
3. The experimental results have conflicts with those in "Deep Neural Networks and Tabular Data: A Survey" by Borisov et al. (October 2021, latest version available at https://ieeexplore.ieee.org/stamp/stamp.jsp?arnumber=9998482). Specifically, the current submission reports maximum accuracy of 76.71% on the covtype dataset, but Table V in the Borisov paper reports over 90% accuracy for eight methods including LightGBM, Catboost, XGBoost, MLP, and SAINT. Accuracy on covtype over 90% for all methods is reported also in Table 2 of "Generalized Boosting" by Suggala et al., NeurIPS 2020. For a fair evaluation, the authors should implement their method in the test harness of Borisov et al. at https://github.com/kathrinse/TabSurvey, which includes hyperparameter optimization.

The following are smaller but not trivial concerns.

What is the motivation for the projection head on top of the decoder? Why not make the decoder produce the reconstructed input directly?

Does the set S of masked coordinates always have fixed cardinality? If masking is independently at random, then normally this would not be the case.

Equation (2) is imprecise because it fails to say in what order to take the elements j of S-bar.

Given that t_mask is a scalar, it has a tiny representational capacity, so why does it need to be learnable?

The L2 loss function in Equation (5) is reasonable only when input values have been scaled, such as by z-scoring.

The dimensionality of the input to the encoder is smaller during pre-training than during inference. Discuss how the transformers handle this.

The maximization in Equation 6 is performed heuristically by gradient ascent in Algorithm 1. Discuss the effects of inexact maximization. Given that epsilon >= 2 and lambda = 1, the non-adversarial loss in Equation 7 may have negligible effect; discuss this.

The toy tabular setting is too simple, with a linear chain dependency structure. Trivial calculation of correlation coefficients, or of cosine similarities would solve this task, as would the method of Klivans and Meka. Show results on a more complex graph structure.

Explain why MET yields large scores, which seem wrong, in the off-diagonal corners of Figure 2.

Adjacent image pixels are highly dependent. Show that MET discovers which pixels are adjacent, after the order of these has been randomized.

Include MET-S in Table 2. The random forest is allowed only 100 trees, too few. This should be a hyperparameter. MET-S has many more hyperparameters than competitor methods, which may be an advantage, even if the data for tuning hyperparameters is strictly separate from test data.

Discuss recent research that compares tabular DL methods systematically. Show results on the same datasets. See Borisov et al., "Revisiting Deep Learning Models for Tabular Data" by Gorishniy et al., and "Why do tree-based models still outperform deep learning on tabular data?" by Grinsztajn et al.

Writing: Some sentences are run-on, with commas instead of semicolons. “et al.” should be followed by a plural verb. In the caption for Figure 1 and before Equation 2, “it’s” should have no apostrophe. Typo: “indentity”.

**Summary Of Ac-Reviewer Meeting:**

No meeting.